# Implementation of basic package of care improved socio-economic conditions of lymphedema patients in rural Mali after two years follow-up

Housseini Dolo[1,2]*, Diadje Tanapo[1], Sekou Oumarou Thera[1], Moussa Sangaré[1,2], Abdoul Fatao Diabaté[1], Lamine Diarra[1], Salif Sériba Doumbia[1], Siaka Yamoussa Coulibaly[1], Ilo Dicko[1], Oumar Coulibaly[1], Ousmane Faye[3], Hamadoun Sangho[2], Seydou Doumbia[2], Yaya Ibrahim Coulibaly[1,3], Thomas B. Nutman[4]

**1** Neglected Tropical Diseases Research and Training Unit, International Center of Excellence in Research, University of Sciences, Techniques and Technologies of Bamako, Bamako, Mali, **2** Department of Teaching and Research in Public Health and Specialties, Faculty of Medicine and Odonto Stomatology of Bamako, University of Sciences, Techniques and Technologies of Bamako, Bamako, Mali, **3** Dermatology Hospital of Bamako, Bamako, Mali, **4** Laboratory of Parasitic Diseases, National Institute of Allergy and Infectious Diseases, National Institutes of Health, Bethesda, Maryland, United States of America

☯ These authros are contributed equally as second co-authors.
* hdolo@icermali.org

## Abstract

Research shows that a local hygiene package washing, wound care, and exercise improves lymphedema (LE) patients' quality of life by reducing ADLA, costs, stigma, and boosting work capacity. The 'LEdoxy' study, a 24-month, multicountry, double-blind, randomized controlled trial of doxycycline plus hygiene, investigated the efficacy of daily doxycycline versus a placebo for lymphatic filariasis in Mali, India, and Sri Lanka. All 'LEdoxy' participants consistently followed a standardized hygiene program. This study, embedded within the previous 'LEdoxy' study, assessed cost reductions in ADLA management and evaluated the social impact of the regular local care based on the HP among LE patients in Mali. We conducted a cross-sectional study embedded in "LEdoxy" study participants in the health districts of Kolondieba and Kolokani from September to December 2021. Questionnaire was used to collect retrospective data on pre-trial conditions and prospective data on post-trial conditions. Data were collected using a digital questionnaire and qualitative methods, including in-depth interviews and focus group discussions with interview guide, without recycling 'LEDoxy' data. These quantitative data were analysed using SPSS version 25.0. We performed a thematic analysis approach using Quirkos version 2. We investigated 196 LE patients with a median age of 56 years [range: 25-70 years]. We observed a reduction in the monthly frequency of acute ADLA from 90.8% (178/196) before the trial to 43.9%, (86/196) after the trial (p<0.05). Importantly, in term of economic evaluation the average ADLA attack management cost decreased significantly

**Data availability statement:** All relevant data are within the paper. However, de-identified data of this study can be requested per the University of Sciences, Techniques and Technologies of Bamako (USTTB) data sharing regulation. Data access requests may be sent to the permanent secretary of the USTTB IRB by sending email to mdiakite@icermali.org or call at 00223 76231191.

**Funding:** This study was funded by the University of Sciences, Techniques and Technologies of Bamako through the research promotion grant to the teaching and research professors awarded to HD. In addition, the study was funded in part (to TBN) by the Division of Intramural Research (DIR) of the National Institute of Allergy and Infectious Diseases, NIH.

**Competing interests:** The authors have declared that no competing interests exist.

from US$20.01(± US $18.56) before the trial to US$5.7(± US $4.78), after the trial (p< 0.001). Patients reported that the hygiene program reduced social isolation and stigma and improved their ability to work. Using a basic care package improved LE patients' quality of life by lowering ADLA costs. To sustain this, community-based programs that include income-generating activities are essential.

## Introduction

Lymphedema (LE) is a chronic swelling of a limb due to an accumulation of lymphatic fluid caused by an obstruction or dysfunction of the lymphatic system [1]. In the context of neglected tropical diseases, it is mainly associated with lymphatic filariasis (LF) [2]. Acute adenolymphangitis attacks (ADLA) is an acute inflammatory episode occurring in pre-existing lymphedema, characterized by bacterial infection of the lymphatic vessels. It manifests as fever, intense pain, skin redness, and sudden worsening of oedema [3].

People living with LE experience ADLA that cause severe pain and loss of productivity [4,5]. Consequently, these individuals are often unable to perform daily activities such as fetching water, harvesting crops, and performing certain household tasks. This contributes to reduced productivity within their community [6,7].

In 2021, the World Health Organization (WHO) estimated that approximately 40 million people suffer from the stigma and disabling clinical manifestations of lymphatic filariasis (LF), of which 15 million have LE [8]. In 2016, a study reported by Dolo et al. identified 557 cases of LE in three previously LF-endemic districts in Bougouni, Kolokani, and Kolondieba [9].

Local hygiene basic package of care is important to reduce oedema progression and prevent associated superinfection that causes ADLA [10,11]. However, ADLA management requires complex care that generates high costs resulting in loss of income for affected individuals and their families who usually are already facing poverty [12,13].

In the LF elimination strategy, WHO advocates burden reduction through the implementation of a basic package of care for LE management. This is one of the main components of the morbidity management and disability prevention program (MMDP), a second pillar of LF elimination as a public health problem [14]. In this context, the Global LF Elimination Program (GPELF) was launched in 2000 to eliminate the disease as a public health problem by 2030 to coordinate prevention and case management activities [14]. In Mali, based on a previous study on LE in three health districts, the majority of people with LE were living in remote rural areas [9].

The WHO recommendation for a basic package of care aligns with the broader global commitment to achieving Universal Health Coverage (UHC). UHC is established on the principles that all individuals and communities should have access to the quality, essential health services they need without suffering financial hardship [15]. Many nations, including Mali, are actively developing strategies to realize this goal. Mali's approach involves a multi-faceted strategy aimed at expanding coverage to its entire population, particularly those in the informal economy and rural areas [15].

In 2018, LeDoxy a randomized trial of doxycycline plus hygiene multicountry study (LEDoxy NCT02927496 registered on 7 Oct 2016) was conducted in Mali, India, and Sri Lanka on the efficacy of doxycycline (compared to placebo) in addition to local hygiene in the early stages of LE [3]. During LEDoxy, all subjects received hygiene kits and training on self-management of LE as recommended by WHO [3]. Patients enrolled were also allowed to consult the nearest health centre gratis when there were any medical needs between two visits of the research team [3]. Thirty-six months after Ledoxy completion, the current study was designed independently to assess the socio-economic impact with an economic evaluation of the costs of managing ADLA of implementing a basic package of care in the management of LE in the two rural study health settings to fill the knowledge gap on the socio-economic impact of the WHO-recommended basic package of care in remote rural communities.

## Methods

### Ethical considerations

This study was approved by the Ethics Committee of the University of Science, Techniques and Technology of Bamako (USTTB), Mali (Approval Number: 2021/231/USTTB). Written informed consent was obtained from all participants prior to enrolment. For participants unable to write, fingerprint consent was obtained in front of a witness in accordance with procedures approved by the Ethics Committee. Participants received full verbal explanations of the study objectives, procedures, risks/benefits, and their right to withdraw without penalty, in the local language (Bambara). All data were anonymized, stored on encrypted servers with password protection, and accessible only to authorized investigators. As compensation, all participants received their transportation costs (a lump sum of 3,000 FCFA equivalent to US $5.25). The protocol was implemented according to the approved version without any deviations.

### Study sites and design

We conducted a cross-sectional study embedded in "LEdoxy" study participants. The same questionnaire was used in the same trial participants to collect retrospective data on their pre-trial conditions and prospective data on post-trial conditions at the same time. Primary data were collected using a digital questionnaire and qualitative methods, including in-depth interviews and focus group discussions guided by a pre-established interview guide, without recycling 'LEDoxy' data. We conducted this study in the health districts of Kolondieba and Kolokani from 26 September to 25 December 2021, using mixed methods (quantitative and qualitative) data collection.

### Study population

The study population consisted of patients of both sexes with lymphedema who had previously been recruited for the LEDoxy clinical trial (NCT02927496) conducted from June 27, 2018, to December 19, 2020. The Kolondieba and Kolokani health districts were selected because of: (i) their documented endemicity of lymphatic filariasis, (ii) their high density of recorded lymphoedema cases, and (iii) the patient follow-up mechanisms put in place as part of the LEDoxy trial [3,9].

The inclusion criteria were as follows: confirmed participation in LEDoxy, residence in the study area for ≥ 6 months prior to data collection, age ≥ 18 years, and voluntary informed consent. The exclusion criteria were as follows: individuals who did not give their consent and participants who had moved outside the study districts.

**Quantitative component sampling.** We used exhaustive sampling by inviting the 220 former LEDoxy participants from both districts. This approach ensured maximum statistical power and representativeness of the initial trial cohort. All eligible patients were contacted and 196 participated (response rate: 89.1%).

**Qualitative component sampling.** Convenience sampling was used to recruit participants. We determined the sample size (8 FGDs; 10 IDIs) based on the principles of data saturation and operational feasibility. Per district, we

conducted: 4 FGDs stratified by gender (men/women) and age group (youth/adults); 5 IDIs with targeted volunteers representing different stages of LE and different socioeconomic backgrounds. A total of eight FGDs and ten IDIs were conducted.

**Questionnaire design and administration.** A digital version of the standardized survey questionnaire was developed and implemented using the Kobo Toolbox platform for electronic data collection via smartphones. Questions related to patients' socio-economic conditions, sustainability of limb hygiene, and aids obtained for this disease management were included. This questionnaire was pre-tested, corrected, and validated by the research team before being used in the field by trained interviewers. Field collected data were checked and edited in real-time by the supervisors and the data manager.

For IDIs and FGDs, interview guides were developed, pre-tested, edited and validated during a workshop in Bamako before being administered in the field. These guides helped interviewers to collect qualitative data. Collected data were reviewed daily by the principal investigator to correct for accuracy and completeness or to address any identified issues to ensure the quality of the collected data.

For the economic evaluation of the data related to the management of one ADLA attack per month, we considered the costs reported by patients. First, the cost of managing one ADLA attack per month was estimated for each patient. Then, the amounts reported by patients were collected to form a dataset. To ensure the reliability of the analysis, extreme values were excluded (a total of 18 data). This exclusion of extreme values enabled us to obtain a more representative and reliable estimate of the average cost of managing one ADLA attack per month.

**Data analyses.** The Statistical Package for Social Sciences (SPSS 25.0) software was used to clean and analyse quantitative data. Categorical variables were summarized as proportion while continuous variables were presented as mean ± standard deviation. Comparisons of proportions were made using Pearson's $Chi^2$ or Fisher's exact test depending on their applicability. For comparison of continuous variables, Wilcoxon signed Rank test was used. Any p values ≤0.05 were considered statistically significant. Adjustments for multiple comparisons were applied using Bonferroni correction. We compared the cost of managing ADLA episode before and 16 months after the local hygiene package implementation as part of LEDoxy. For qualitative data, records were captured and coded as themes and subthemes, and Quirkos software version 2 was used to support data analysis.

## Inclusivity in global research

Additional information regarding the ethical, cultural, and scientific considerations specific to inclusivity in global research is included in the r Information (S1 Checklist).

## Results

### Quantitative results

**Sociodemographic characteristics of LE patients after LEDoxy.** A total of 196 patients were included in this study. The health district of Kolondieba had the largest number of participants with 72.4% (142/196). Females represented 86.7% (170/196) of the participants. The median age of the participants was 56 years [25–70]. More than half of the patients were married (59.7% (117/196)); most of them were not educated 90.8% (178/196), and 79.1% (155/196) were farmers (Table 1).

### LE-related difficulties and regularity of limb washing 16 months after LEDoxy

The proportions of patients who reported not having received help from a relative or friend in the village for the management of ADLA crises in Kolokani and Kolondieba health districts were 94.4% (51/54) and 88% (125/142) respectively. The proportions of patients who did not experience LE-related stigma or discrimination after LEDoxy study in Kolokani and

**Table 1. Description of patients affected by LE 16 months after LEDoxy in the health districts of Kolondieba and Kolokani in 2021.**

| Socio-demographic characteristics | Health districts | | Total |
|---|---|---|---|
| | Kolokani | Kolondieba | |
| | n = 54 | n = 142 | N = 196 |
| | n (%) | n (%) | n (%) |
| **Gender** | | | |
| Female | 43 (79.6) | 127 (89.4) | 170 (86.7) |
| Male | 11 (20.4) | 15 (10.6) | 26 (13.3) |
| **Age group** | | | |
| Under 56 years old | 21 (38.9) | 76 (53.5) | 97 (49.5) |
| 56 years old and above | 33 (61.1) | 66 (46.5) | 99 (50.5) |
| Median age [Min-Max] | 60 years [30–70] | 54 years [25–70] | 56 years [25–70] |
| **Level of education** | | | |
| Schooled | 4 (7.4) | 14 (9.9) | 18 (9.2) |
| Unschooled | 50 (92.6) | 128 (90.1) | 178 (90.8) |
| **Marital status** | | | |
| Single | 1 (1.8) | 4 (2.8) | 5 (2.5) |
| Married | 34 (63) | 83 (58.5) | 117 (59.7) |
| Widow(er) | 19 (35.2) | 55 (38.7) | 74 (37.8) |
| **Profession** | | | |
| Shopkeeper | 1 (1.9) | 4 (2.8) | 5 (2.6) |
| Cultivator | 39 (72.2) | 116 (81.7) | 155 (79.1) |
| Teacher | 0 (0) | 1 (0.7) | 1 (0.5) |
| Housekeeper | 11 (20.4) | 13 (9.2) | 24 (12.2) |
| Other | 3 (5.5) | 8 (5.6) | 11 (5.6) |

Min-Max: Minium-Maximum.

Kolondieba health districts were 94.4% (51/54) and 82.4% (117/142) respectively. The proportions of patients reporting continued limb washing after LEDoxy in the two districts were 78.2% (111/142) in Kolondieba and 83% (45/54) in Kolokani (Table 2).

## Changes in ADLA management cost

The average cost of managing an ADLA attack decreased significantly from 13,481 FCFA (±10,456 FCFA) equivalent to US $20.01(± US $18.56) before the introduction of basic package of care to 3,843 FCFA (±2,692 FCFA) equivalent to US $5.7 (± US $4.78) after its implementation for 2 years (p < 10–3) (Fig 1).

## Qualitative results

### Lymphedema patients' productivity

LE can have a significant impact on the productivity of affected individuals. Swelling of the arms or legs can make it difficult to perform daily duties. Treatment of LE often involves the use of compression bandages, which can help reduce swelling and improve mobility. However, this condition can still result in missed workdays and reduced productivity. This can have a negative impact on the socio-economic status of those affected and their families. The main themes that were discussed were the inability to work, particularly during episodes of ADLA.

**Table 2.** Frequencies of social, support, and hygiene practices related constraints within LE-affected patients in the two health districts in 2021.

| LE-related difficulties and regularity of limb washing | Health districts | | p |
|---|---|---|---|
| | Kolokani | Kolondieba | |
| | n (%) | n (%) | |
| **Received support/ assistance for LE** | | | |
| No | 51 (94.4) | 125 (88) | 0.29 |
| Yes | 3 (5.6) | 17 (12) | |
| **Social difficulties related to LE** | | | |
| No | 51 (94.4) | 117 (82.4) | 0.03 |
| Yes | 3 (5.6) | 25 (17.6) | |
| **Continuity of limb washing practices** | | | |
| No | 9 (16.7) | 31 (21.8) | 0.55 |
| Yes | 45 (83.3) | 111 (78.2) | |

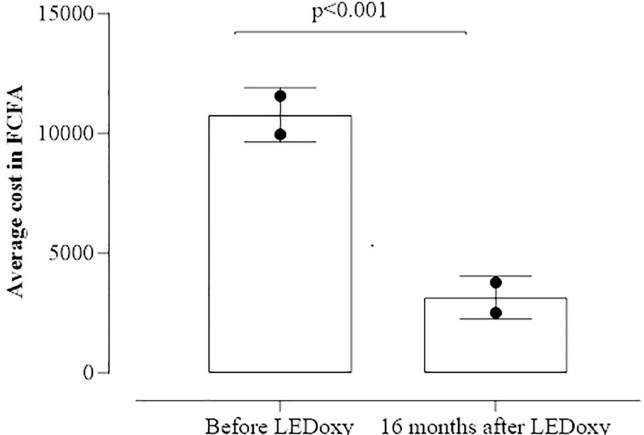

**Fig 1. Reduction on the average expenses for ADLA crisis management within lymphedema patients before and after the use of hygiene measures package in the health districts of Kolondieba and Kolokani in 2021.**

A woman explained that she could not do any expected farming or other field work and that this necessarily resulted in a loss of income.

"*Our main source of income is farming and gardening; this disease prevents us from carrying out these different activities. So, we have almost no other source of income.*" [FGD with elderly women in Kebila, Kolondieba]

Another participant said that before her disability, she used to farm and do small-scale horticulture, but with the acute crises, this is no longer possible.

"*I used to grow peanuts to earn money, but now I can't do it because of the acute crises*". [IDI with elderly women in Kebila, Kolondieba].

Patients with LE mentioned working hard, walking long distances, and being in contact with water, especially in winter, as factors that induced ADLA. This is illustrated by the following quote:

"*During the study, things were much better because we were able to work during the rainy season, but since the study was stopped, it has become difficult, if not impossible, again*" [FGD with elderly men from Kebila, Kolondieba].

Some participants reported that they were less productive than peers of their age group in the village. For example, a woman during an interview stated that:

"*With this disease, I can no longer produce shea butter or néré powder of "Parkia biglobosa" while all my age mates in the village do it to earn money.*" [IDI with a woman from Kolondieba].

A group of LE carriers stated that the economic impact does not affect only the patient but also their close relatives.

"*I don't work anymore and sometimes I find it hard to ask the children for money. Sometimes, I used to take loans from people in order to pay for my medicine during crises, but, since LEDoxy, I can go anywhere and work.*" [FGD with elderly men in Sabougou, Kolokani]

### Social impact of LE on patients

LE can led to social isolation, and the use of hygiene measures can help reduce the stigma associated with this condition. Thus, a man in Kolondieba, during a focus group discussion, said: "*Some people feel disgusted with sick people like us and, except for a few, no one really cares about us*". [FGD with elderly men in Kolondieba].

Another woman, during an individual interview, stated that:

"*Yes, even now people in my family think that I refuse to be a housekeeper, to prepare for the family, whereas it is very difficult for me to stand up for long periods of time*". [IDI with a woman from Kolondieba]

The use of hygiene measures has enabled some people to participate again to their communities' collective works. This is illustrated by a quote from a woman interviewed during a focus group discussion:

"*We used to have difficulty walking, but thanks to the LEDoxy study, I can walk long distances, and I can take part in collective activities*" [FGD with elderly women from Kolondieba].

### Discussion

LE is one of the complications of LF, which is an NTD affecting many people in rural, remote, and hard-to-reach areas of Mali. The current study participants' median age was 56 years [range: 25–70 years]. In rural Mali, where subsistence agriculture dominates, individuals aged 25–70 years typically remain economically active regardless of retirement age. This age group is more frequently affected by LE because this chronic complication occurs many years after infection, and once it has occurred, it is almost irreversible without early and permanent local hygiene or a reversal of the primary cause that could sometimes be resolved using lymphatic surgery [16]. The high number of cases in this age group may be due to the beneficial effects of mass treatment with ivermectin and albendazole that protected younger age groups by reducing their exposure to LF parasites and therefore their likelihood to develop LE [17]. The MDA was instituted in Kolondieba from 2005 and in Kolokani from 2006. This result may help draw the attention of political and financial decision-makers to the need to target the morbidity management and disability prevention (MMDP) efforts at these at-risk age groups. Most of the participants were farmers, 78.1%. This means that LE in Mali should be considered in rural areas as a serious threat to patients' productivity. In these areas, agriculture is the basis of subsistence and therefore pivotal to the local economy.

In our sample, 90.8% (178/196) of participants had received no formal education (Table 1), reflecting the national reality where the net enrolment rate in rural areas was 68.3% for primary education in 2020 [18]. This implies that special attention must be paid to schooling for people living in rural areas to promote health education and improve the prevention and self-management of ADLA crises in these areas. These results provided valuable information on the socio-demographic characteristics of people affected by LE, which may help to develop tailored treatments and therapeutic approaches especially when it comes to sensitization and training materials developments since those should consider features such as the very low literacy rate.

As part of the LEDoxy study, LE carriers were trained in local hygiene use and provided with free-of-charge treatment at community health centres and, for the intervention group. The average cost of managing an ADLA crisis decreased significantly 16 months after LEDoxy. This suggests that washing limbs daily for a prolonged period reduced the frequency and severity of acute seizure episodes, resulting in lower management costs. The average monthly income of people living in these rural areas of Mali is about US $ 105.82 equivalent to 64,485 F CFA [19]. The use of a basic package of care is a low-cost alternative for the prevention of ADLA crises. The use of this package reduces the frequency of attacks and therefore decreases the cost of LE management while the loss of time of work also decreases. Given the significant benefits of affected limb hygiene, it is important for LF elimination programs to implement and find ways to sustain this hygiene measure within LE patients for a better and effective prevention/management of LE [6]. To do so, communicating about the economic burden of LE would be useful to convince donors to support LF elimination programs for patients and their relatives in these poor environments. Our findings on the economic impact of the basic care package have profound implications for the advancement of Universal Health Coverage (UHC) in Mali. A core objective of UHC is to provide financial protection by minimizing out-of-pocket health expenditures that can be catastrophic for low-income households [20]. This study demonstrated a significant decrease in the average cost of managing an ADLA attack, from US$20.01 to US$5.7. For a rural population where most participants are subsistence farmers and the average monthly income is approximately US$105.82, this reduction represents a substantial relief from a crippling financial burden. The high pre-intervention costs are a classic example of the type of health-related expenditure that UHC aims to eliminate. Therefore, our results provide strong evidence that integrating this low-cost hygiene intervention into a national health strategy is a powerful and cost-effective tool for enhancing financial risk protection, a cornerstone of UHC.

The morbidity management component through the implementation of community-based morbidity management programs would also be necessary to support the use of the basic package of care and further reduce the burden of disease. In addition, the WHO recommends further improvements for health facilities, such as ensuring the availability of antibiotic creams and tablets in health facilities located in endemic areas. In addition, it is essential to ensure the presence of qualified staff for the effective management of lymphedema cases in general, and ADLA attacks in particular. These measures strengthen the capacity of health facilities to respond to the specific medical needs of people affected by LE.

Some patients reported a positive effect of hygiene measures on their productivity. The costs associated with managing the ADLA account for the bulk of the expenditure, and a large part of the economic burden is due to the indirect costs associated with the loss of economic productivity [13]. LE and ADLA may require patients to work fewer days per year or fewer hours per day, and they may have lower earnings because they cannot engage in work which may increase the risk of acute attacks [7,8]. In advanced stages, LE is very disabling [21]. In the intermediate stages, it results in partial disability with considerable loss of productivity, and the early stage progresses to the advanced stage in the absence of adequate management [13]. Rigorous skin hygiene is essential to avoid superinfection, as the affected limb's skin is more vulnerable to infection [22]. The basic package of care is effective in reducing the frequency and intensity of ADLA and thus helping to increase patients' productivity. Current results imply that hygiene measures are essential to mitigate the economic impact of LE, by reducing work absences and productivity losses. Therefore, policymakers can be advised to promote the implementation of LE management programs based on hygiene measures to reduce the incidence of ADLA episodes and improve the quality of life (QoL) of LE patients. Additionally, stakeholders can also provide increased

support for operational research to assess the effectiveness and feasibility of other LE management interventions, such as self-care protocols and strategies for integrating hygiene measures into the minimum package of activities of community health centres to improve LE management in remote LE-affected communities.

In a focus group discussion, patients reported a feeling of social isolation related to LE. After the introduction of the basic package of care, more than half of the patients (76.5%) reported that they did not experience any social difficulties related to the condition. The physical, social, and psychological suffering caused by LE is considerable [10]. The reduction of social difficulties using local hygiene measures is evidence of the effectiveness of this intervention [11,12].

LE patients in rural areas of Mali have few income-generating activities and most of them are older persons. Social difficulties such as stigma and reduced ability to perform daily activities lead to social isolation, which in turn contributes to depression and other health issues [13]. In addition, an increase in the incidence of ADLA can hurt socioeconomic status due to work disability [14]. This result highlights a significant decrease in social difficulties related to LE with regular use of local hygiene measures. However, further research is needed including studies testing large-scale interventions to develop a better understanding of the impact of the WHO-recommended basic package of care including local hygiene measures, physical exercise, antibiotic and antifungal topical use, and wound avoidance, to see how they will lead to improved quality of life and under what circumstances this care may be easiest. This requires an assessment of QoL based on a broader perspective including environmental determinants as well as the health centre frequented by patients in the context of continuing education of rural health professionals and LE patients. Large-scale initiatives to reduce the social difficulties associated with LE must also be undertaken. This requires a clearer understanding of the process of occurrence of these difficulties faced by patients so that more effective interventions can be implemented. Social support by the community, local healthcare workers' assistance, and the donation of hygiene kits to patients with LE could significantly and sustainably decrease the social difficulties associated with this condition.

Some participants indicated that they no longer washed their affected limbs. According to qualitative data collected during individual and group interviews, in the absence of regular limb hygiene, patients resort to traditional remedies that are sometimes ineffective and costly. The practices reported included: applying herbal preparations to the affected limb, making incisions in the skin to 'drain the edema,' and purchasing painkillers from street vendors. The basic package of care is an effective and less costly means for LE management [15]. If all affected patients practice hygiene according to WHO recommendations, there would be significant benefits in terms of health, QoL, and economic productivity [16]. To achieve this, a method based on the integration of LE management into the current health system and the organization of patients into self-support associations for improved follow-up care must be undertaken [23]. Doing so would require local readaptation and tailoring that would consider local features and culture.

In rural Mali, the main activity is agriculture, which is highly impacted by the LE and related ADLA, so an effective and efficient program to manage LE must include the establishment of income-generating activities (IGAs), the types and modalities of which must be discussed and thought out by associations of people living with LE and by the community to make this action sustainable.

This mixed-methods study provides data exclusively from rural areas where lymphatic filariasis was endemic, based on primary data collection. Quantitative rigor (n = 196, or 89.1% of the initial cohort) was complemented by rich qualitative information (8 group discussions, 10 individual interviews).

However, the study has some limitations. First, non-probability sampling (exhaustive for quantitative components, convenience sampling for qualitative components) may limit generalizability. Second, the results apply specifically to similar rural contexts where LE is endemic, but not to urban contexts. Third, the use of the guaranteed minimum wage as an economic reference was not optimal for agricultural populations (79.1%). Finally, memory bias may affect self-reported socioeconomic data before and after the intervention. Despite sampling constraints, the study offers rare longitudinal socio-economic data in a neglected tropical disease context.

## Conclusion

This study suggests that the use of a basic package of care significantly improved the socioeconomic conditions of LE carriers mainly by reducing the cost management of ADLA attacks. There is a need to establish a community-based lymphedema management program with income-generating activities as a strategy to support the sustainability of these measures.

## Supporting information

**S1 Checklist. Inclusivity in global research.**
(DOCX)

## Acknowledgments

We would like to thank the National Lymphatic Filariasis Elimination Program for its collaboration in implementing this study. We thank the health districts officers of Kolondieba and Kolokani for making it possible to carry out this study as well as the community health workers. We thank Ledoxy study team for sharing the participants data to support the embedded current study.

## Author contributions

**Conceptualization:** Housseini Dolo, Diadje Tanapo, Sekou Oumarou Thera, Seydou Doumbia, Yaya Ibrahim Coulibaly.

**Data curation:** Lamine Diarra, Oumar Coulibaly.

**Formal analysis:** Housseini Dolo, Diadje Tanapo, Sekou Oumarou Thera, Abdoul Fatao Diabaté, Salif Seriba Doumbia, Ilo Dicko.

**Funding acquisition:** Housseini Dolo.

**Investigation:** Diadje Tanapo, Sekou Oumarou Thera, Moussa Sangare, Abdoul Fatao Diabaté, Lamine Diarra, Salif Seriba Doumbia, Siaka Yamoussa Coulibaly, Oumar Coulibaly.

**Methodology:** Housseini Dolo, Diadje Tanapo, Sekou Oumarou Thera.

**Project administration:** Housseini Dolo.

**Resources:** Housseini Dolo, Ousmane Faye, Hamadoun Sangho, Thomas B. Nutman.

**Software:** Oumar Coulibaly.

**Supervision:** Diadje Tanapo, Sekou Oumarou Thera, Moussa Sangare, Abdoul Fatao Diabaté, Lamine Diarra, Salif Seriba Doumbia, Siaka Yamoussa Coulibaly, Ilo Dicko.

**Validation:** Housseini Dolo, Hamadoun Sangho, Seydou Doumbia, Yaya Ibrahim Coulibaly, Thomas B. Nutman.

**Visualization:** Housseini Dolo, Siaka Yamoussa Coulibaly.

**Writing – original draft:** Housseini Dolo, Diadje Tanapo, Sekou Oumarou Thera, Moussa Sangare, Abdoul Fatao Diabaté, Ilo Dicko.

**Writing – review & editing:** Housseini Dolo, Ousmane Faye, Hamadoun Sangho, Seydou Doumbia, Yaya Ibrahim Coulibaly, Thomas B. Nutman.

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
