## [Decision Letter · Decision Letter 0]

5 Jun 2025

PGPH-D-25-00707

Implementation of Basic Package of Care Improved Socio-Economic Conditions of Lymphedema Patients in rural Mali after two years follow-up.

Dear Dr. Dolo,

Thank you for submitting your manuscript to PLOS Global Public Health. After careful consideration, we feel that it has merit but does not fully meet PLOS Global Public Health’s publication criteria as it currently stands. Therefore, we invite you to submit a revised version of the manuscript that addresses the points raised during the review process.

Congratulations to the authors for this very important manuscript which clearly depicts the importance of essential package of care in reducing the acute ADL attacks and thereby improving the quality of life of patients with chronic lymphedema and the improvement in socioeconomic situation and study itself will help this to be an important advocacy point for continuing MMDP and arriving at GPELF goalsEven though Ledoxy study was mentioned, there seems to be agap to understand the relation between the two studies. So a short summary of the study will help readers to understand this well.It would be good to mention about the Universal Health coverage in the context of patient care and self care. The points raised by reviewers also should be addressed and other points like mentionng as the acute seizure episodes

P

We look forward to receiving your revised manuscript.

Kind regards,

Suma Krishnasastry, MBBS, MD,DNB, FRCP (Edin)

Academic Editor

Journal Requirements:

Additional Editor Comments (if provided):

Reviewers' comments:

Reviewer's Responses to Questions

**Comments to the Author**

1. Does this manuscript meet PLOS Global Public Health’s publication criteria?

Reviewer #1: Yes

Reviewer #2: Yes

Reviewer #3: Yes

Reviewer #4: Yes

2. Has the statistical analysis been performed appropriately and rigorously?

Reviewer #1: Yes

Reviewer #2: Yes

Reviewer #3: No

Reviewer #4: Yes

3. Have the authors made all data underlying the findings in their manuscript fully available (please refer to the Data Availability Statement at the start of the manuscript PDF file)?

Reviewer #1: Yes

Reviewer #2: Yes

Reviewer #3: Yes

Reviewer #4: Yes

4. Is the manuscript presented in an intelligible fashion and written in standard English?

Reviewer #1: Yes

Reviewer #2: Yes

Reviewer #3: Yes

Reviewer #4: Yes

Reviewer #1: Abstract:

• The abstract needs to include 1-2 sentences providing global context about LF/LE burden, particularly in Sub-Saharan Africa, to better frame the study's importance.

• Key qualitative findings about stigma reduction and improved work productivity should be added to the abstract's results section to fully represent the study's outcomes.

Introduction:

• The rationale needs strengthening by incorporating recent data (2020-2024) on LE's socio-economic impact in Africa to better justify the study's timeliness.

• The novelty of this research should be explicitly stated, particularly how it fills knowledge gaps

Methods:

• The choice of cross-sectional "before-and-after" design requires justification, especially if longitudinal data was available from the LEDoxy trial.

Discussion:

• The discussion should include comparisons with similar programs in other countries to contextualize the findings.

Reviewer #2: This is a good and well written manuscript which meets the standards for publication. However, a few sections have been highlighted for your attention. Kindly click on the highlighted portions to reveal the comments to be addressed.

Reviewer #3: Overall Impression

This manuscript " Implementation of Basic Package of Care Improved Socio-Economic Conditions of Lymphedema Patients in rural Mali after two years follow-up" addresses a highly pertinent public health issue; the socio-economic impact of lymphedema (LE) and the effectiveness of the Basic Package of Care (BPC) in rural, endemic settings. The study's findings, highlighting a substantial reduction in acute filarial attacks (ADLA) frequency and management costs, coupled with positive reports on reduced social isolation, stigma, and improved productivity, are compelling and advocate strongly for the sustained implementation of these care programs. The mixed-methods approach is a notable strength, offering both quantitative evidence and invaluable qualitative insights into patients' lived experiences.

However, the manuscript, in its current form, requires significant revisions to enhance methodological clarity, analytical rigor, and adherence to scholarly conventions. The abstract's description of the study design is imprecise and needs to be clarified. Details regarding data collection timelines and the handling of outliers should be explicitly stated for transparency.

Furthermore, the discussion section, while insightful, needs more critical engagement with the literature. Claims, particularly concerning age group prevalence, irreversibility of LE, and the role of traditional practices, require more robust and accurately cited evidence. Consistency in citation style and precise statistical notation are also essential. The organization of the discussion, with its fragmented subheadings, could be improved by transitioning to a more fluid, paragraph-based thematic flow. Addressing these points will significantly strengthen the manuscript's scientific merit and impact.

Specific Comments

Abstract

Line 27 to Line 28: A “local hygiene package”? and “quality of life”? Can you be more specific? Especially for the quality of life, which aspects?

Line 30 to Line 32: A “cross-sectional study” was used, but the authors mentioned “before and after approach” which is pointing towards a longitudinal design with two data collection points rather than cross-sectional that measures one data point. The title suggests this is a follow-up study (may also be a cohort design), or was this an interventional study (quasi experimental design)? Please make it clearer. Also, was the data purely primary, secondary, or a combination of both?

Line 32 to Line 35: Mention the time period when the study was done.

Line 36: 196 LE patients were enrolled “median 56 years [27-70] years” is what? An interquartile range or range?

Line 38: and Line 40: (p<10-3)? Is this a P-Value? If so, edit accordingly (example p < 0.05 is a more standard way of writing it)

Line 44 to Line 45: This sentence is not clear. Please make clearer.

Main Article

Introduction

Ln 49 and Ln 50: Lymphedema (LE) and Acute adenolymphangititis (ADLA) introduction is good following a funnel approach.

Ln 68 – Ln 70. Maintain consistency with citation at the end of the sentence as done previously.

Ln 74 – Ln 76: “During the LEDoxy study…” in-text citation only included WHO (14), (13) was not included.

Line 76 to Line 81: Was this still talking about the LEDoxy study? Please add the citation and make this clearer.

Line 81: The font looks smaller/different, adjust to match the rest if the article.

Methods

Line 83 to Line 89: Started with ethical consideration. Consider following the journal guidelines for the arrangement.

Line 91: Please refer to the previous comment on line 30 to line 32 of the abstract.

Line 96 to Line 107: For the study population consider mentioning the age range and sex of the study population. The sampling frame is also not clear; is the sampling frame the health districts or the LE patients? Is so, did the authors have any exclusion criteria? Write about the sampling for the quantitative and qualitative data collection on separate paragraphs for clarity. Elaborate on the sample size calculation and the sampling technique and procedure. Description of the sampling technique and procedure is not clear.

Line 107: ‘A total of eight FGDs and ten IDIs were planned’ This has been done already please edit.

Questionnaire design and administration

Line 109 to Line 110: “conceived”. Rephrase sentence that may be confusing. I understand that an electronic version of the survey questionnaire was used for data collection. And based on line 112 to line 113, the questionnaire was standardized through the process of pre-testing, correction, and validation. Also interviewers were trained. It is safe to edit line 109 to line 110 to capture the fact that an electronic version of the standardized survey questionnaire was used to collect data. (Add clarity to this)

Line 114: mentions kobotoolbox was used. Mentioning this once is sufficient.

Line 121: “Economic Evaluation” was not captured in the abstract or the objectives. Please check to ensure it is captured in all sections.

Data analysis

Ln 134: “p value =<0.05…” use the appropriate symbol (example ≤).

Results

Table 1: remove the bold horizontal borderlines separating the variables of the sociodemographic characteristics. Leave only the borderline for the headings and after the last row.

Table 2: see comment for table 1.

Figure 1 is tiny and not visible. Adjust.

Generally for the results, I suggest you should create major subheadings for the presentation of quantitative results and for the qualitative results to guide the reader.

Discussion

Line 220 to Line 221: median age… also indicate if [25-70] is range or interquartile range?

Line 221: “This age group corresponds to the professionally active age group (1)”. The statement may not be valid. I checked the reference; it is an article in French as such could not verify. Please check. What is the age of retirement in Mali? I was able to see age 60 years and below, please confirm.

Line 222 to Line 225: maintain the consistency of the in-text citations at the end like it was done in most of the article. The reference no. 2. In the reference list refers to the abstract of the study not the full text. The full text (Pfarr KM, Debrah AY, Specht S, Hoerauf A. Filariasis and lymphoedema. Parasite Immunol. 2009 Nov;31(11):664-72. doi: 10.1111/j.1365-3024.2009.01133.x. PMID: 19825106; PMCID: PMC2784903.) does not support the claims in the authors’ sentence “This age group is more frequently affected by LE because this chronic complication occurs many years after infection, and once it has occurred, it is almost irreversible without early and permanent local hygiene (2) or a reversal of the primary cause that could sometimes be resolved using lymphatic surgery.” Please check and edit your citation.

Line 225 to Line 227: “The high number of cases in this age group may be due to the beneficial effects of mass treatment with ivermectin and albendazole that protected younger age groups by reducing their exposure to LF parasites and therefore their likelihood to develop LE.” Please provide citation.

Line 233 to Line 234: “In the Malian context, the schooling rate is usually low in rural areas.” Is this referring to findings from the study? Please make it clearer.

Line 270: “In advanced stages, LE is very disabling” Provide citation for this.

Line 311: starts with small letter “s” please edit.

Line 311 to Line 315: “In the absence of regular limb hygiene, patients resort to remedies and treatments that can be ineffective and costly. In the Malian context, traditional practices for the management of LE include herbal preparations applied to the affected limb, skin incision, and analgesics purchased from itinerant drug vendors outside of pharmacies.” Provide citation if these are not findings from your study.

Limitations

Elaborate on the limitations in design; your sample size, sampling (if probability sampling was not used), also mention if your results are not generalizable at a National or district level as a limitation. Also mention the strengths of your study before the limitation.

I suggest the subheadings (Cost of LE management, productivity of LE patients, social impact of lymphedema, LE hygiene practice sustainability, and Limitations) in the discussion should be eliminated. Start discussion a new idea or theme on a new paragraph, this will improve the flow.

Generally, check your references ensure they are all in English as you submitted your manuscript in English.

Reviewer #4: Socio-economic impact of lymphatic filariasis(LF) should be clearly mentioned.

Rationale of the study should be more specific

Results should be more elaborative

Statistical analysis was done appropriately

**Do you want your identity to be public for this peer review?** For information about this choice, including consent withdrawal, please see our Privacy Policy

Reviewer #1: No

Reviewer #2: No

Reviewer #3: **Yes: ** Suleiman Idris Ahmad

Reviewer #4: **Yes: ** MUHAMMED SYEDUL ALAM

---

## [Decision Letter · Decision Letter 1]

28 Aug 2025

PGPH-D-25-00707R1

Implementation of Basic Package of Care Improved Socio-Economic Conditions of Lymphedema Patients in rural Mali after two years follow-up.

Dear Dr. Dolo,

Thank you for submitting your manuscript to PLOS Global Public Health. After careful consideration, we feel that it has merit but does not fully meet PLOS Global Public Health’s publication criteria as it currently stands. Therefore, we invite you to submit a revised version of the manuscript that addresses the points raised during the review process.

Reviewer 1 has requested some final revisions before we can accept your submission for publication in PLOS One. Please carefully address all of their comments below. 

We look forward to receiving your revised manuscript.

Kind regards,

Jennifer Tucker, PhD

Staff Editor

Journal Requirements:

Additional Editor Comments (if provided):

Reviewers' comments:

Reviewer's Responses to Questions

**Comments to the Author**

Reviewer #1: All comments have been addressed

Reviewer #3: All comments have been addressed

Reviewer #4: All comments have been addressed

publication criteria?

Reviewer #1: Yes

Reviewer #3: Yes

Reviewer #4: Yes

3. Has the statistical analysis been performed appropriately and rigorously?

Reviewer #1: Yes

Reviewer #3: Yes

Reviewer #4: Yes

4. Have the authors made all data underlying the findings in their manuscript fully available (please refer to the Data Availability Statement at the start of the manuscript PDF file)?

Reviewer #1: Yes

Reviewer #3: Yes

Reviewer #4: Yes

5. Is the manuscript presented in an intelligible fashion and written in standard English?

Reviewer #1: Yes

Reviewer #3: Yes

Reviewer #4: Yes

Reviewer #1: The authors have made substantial revisions that address prior concerns, resulting in a manuscript that now meets PLOS Global Public Health criteria. The study is methodologically sound, ethically rigorous, and its conclusions are well-supported by the data. The mixed-methods approach strengthens validity, and the design—cross-sectional with retrospective and prospective elements—is now clearly described. Minor refinements are suggested: in the Abstract, explicitly note the economic evaluation (e.g., “assessed cost reductions in ADLA management”); in the Introduction, briefly describe LEDoxy as a randomized trial of doxycycline plus hygiene to contextualize the socio-economic focus; and in the Methods, clarify whether adjustments for multiple comparisons were applied. The Data Availability Statement complies with policy, but the authors should specify whether de-identified data will be shared unconditionally or require ethics committee approval. Language edits include changing “costs of car” to “costs of care” and condensing a sentence in the Discussion to: “In rural Mali, where subsistence agriculture dominates, individuals aged 25–70 typically remain economically active regardless of retirement age.” The Limitations section could be strengthened by noting that, despite sampling constraints, the study offers rare longitudinal socio-economic data in a neglected tropical disease context. Finally, ensure all references are accessible and appropriately updated. Overall, the revisions have significantly improved the clarity, rigor, and contribution of this work.

Reviewer #3: It was a great pleasure reviewing the revised version of your manuscript. The manuscript was extensively edited. My comments have been addressed.

Reviewer #4: All comments have been adequately addressed, written in standard English, meets publication criteria, all data fully available.

**Do you want your identity to be public for this peer review?** For information about this choice, including consent withdrawal, please see our Privacy Policy

Reviewer #1: No

Reviewer #3: **Yes: ** Suleiman Idris Ahmad

Reviewer #4: **Yes: ** MUHAMMED SYEDUL ALAM

---

## [Decision Letter · Decision Letter 2]

28 Oct 2025

Implementation of Basic Package of Care Improved Socio-Economic Conditions of Lymphedema Patients in rural Mali after two years follow-up.

PGPH-D-25-00707R2

Dear Dr Dolo,

We are pleased to inform you that your manuscript 'Implementation of Basic Package of Care Improved Socio-Economic Conditions of Lymphedema Patients in rural Mali after two years follow-up.' has been provisionally accepted for publication in PLOS Global Public Health.

Best regards,

Julia Robinson

Executive Editor

Reviewer Comments (if any, and for reference):

Reviewer's Responses to Questions

**Comments to the Author**

Reviewer #1: All comments have been addressed

Reviewer #4: All comments have been addressed

publication criteria?

Reviewer #1: Yes

Reviewer #4: Yes

3. Has the statistical analysis been performed appropriately and rigorously?

Reviewer #1: Yes

Reviewer #4: Yes

4. Have the authors made all data underlying the findings in their manuscript fully available (please refer to the Data Availability Statement at the start of the manuscript PDF file)?

Reviewer #1: Yes

Reviewer #4: Yes

5. Is the manuscript presented in an intelligible fashion and written in standard English?

Reviewer #1: Yes

Reviewer #4: Yes

Reviewer #1: (No Response)

Reviewer #4: All comments addressed properly

**Do you want your identity to be public for this peer review?** For information about this choice, including consent withdrawal, please see our Privacy Policy

Reviewer #1: No

Reviewer #4: **Yes: ** MUHAMMED SYEDUL ALAM
